# Non-Destructive Testing of Technical Conditions of RC Industrial Tall Chimneys Subjected to High Temperature

**DOI:** 10.3390/ma12122027

**Published:** 2019-06-24

**Authors:** Marek Maj, Andrzej Ubysz, Hala Hammadeh, Farzat Askifi

**Affiliations:** 1Faculty of Civil Engineering, Wroclaw University of Science and Technology, Wroclaw 50-370, Poland; andrzej.ubysz@pwr.edu.pl; 2Faculty of Engineering, Middle East University, Amman 11831, Jordan; hhamm2131@gmail.com; 3Department of Structure Engineering, Faculty of Civil Engineering, Damascus University, Damascus PO Box 30621, Syria; frzataskifi@gmail.com

**Keywords:** nondestructive testing, thermography, monitoring of structures, reinforced concrete chimney, corrosion processes, service life of a structure

## Abstract

Non-destructive tests of reinforced concrete chimneys, especially high ones, are an important element in assessing their condition, making it possible to forecast their safe life. Industrial chimneys are often exposed to the strong action of acidic substances, They are negatively exposed to the condensation of the flue gases. Condensate affects the inside of the thermal insulation and penetrates the chimney wall from the outside. This is one reason for the corrosion of concrete and reinforcing steel. Wet thermal insulation settles, and drastically reduces its insulating properties. This leads to an increase in temperature in the reinforced concrete chimney wall and creates additional large variations in temperature fields. This consequently causes a large increase in internal forces, which mainly increase tensile and shear stresses. This results in the appearance of additional cracks in the wall. The acid condensate penetrates these cracks, destroying the concrete cover and reinforcement. Thermographic studies are very helpful in monitoring the changes in temperature and consequently, the risk of concrete and reinforcement corrosion. This simple implication between changes in temperature of the chimney wall and increasing inner forces as shown in this article is particularly important when the chimney cannot be switched off due to the nature of the production process. Methods for interpreting the results of thermovision tests are presented to determine the safety and durability of industrial chimneys.

## 1. Introduction

Industrial reinforced concrete chimneys are often exposed to a chemically aggressive environment. The combustion gases conveyed via the chimney undergo condensation inside it or are dissolved in precipitation, becoming strongly acidic liquids. The concrete/acid contact results in the corrosion of the concrete and after the concrete cover is penetrated, in the corrosion of the steel also. The two corrosion processes result in the rapid degradation of the chimney structure.

As regards chimneys, one should bear in mind not only the high cost and the technological challenges involved in their construction but also that they perform an essential role in the production processes and so cannot be put out of service for repairs. For example, in steelworks and coking plants, the damping of the furnace from which the combustion gases are conveyed to a chimney results in the destruction of the whole power unit. Therefore, nondestructive tests are vital for both assessing the current technical condition of the chimney and monitoring the degradation processes over its whole service life [1].

## 2. Causes and Effects of Reduced Effectiveness of Chimney Thermal Insulation

One of the major causes of the degradation of the chimney’s reinforced concrete (RC) shell is the too rapid fall in the temperature of the combustion gas as it flows through the chimney flue. The chimney wall consists of the following three layers:A reinforced concrete shaft;An internal wall (e.g., made of fire brick) constituting the chimney’s inner lining which is very resistant to high temperatures;A mineral wool layer, placed between the two walls, serving as thermal insulation.

The durability of a chimney to a considerable degree depends on the quality and longevity of the thermal insulation. Since the chimney cannot be taken out of the production process, it is highly important to constantly monitor the condition of the insulation. Particularly suitable for this purpose are non-destructive testing methods. If the RC shell were removed in places to expose the insulation, the places could become corrosion centres and a thermal bridge could form.

When designing the geometry and thermal insulation of a chimney, one should consider the combustion gas inlet and outlet temperature, the amount and rate of flow of the flue gas and its condensation temperature. Significant internal forces can also arise from a temperature difference [2].

The above parameters are determined by

The amount of the exhausted gas;Changes in flue gas temperature due to changes in production technology;The daily changes of the physicochemical parameters of the chimney environment, such as the wind velocity, the atmospheric pressure, the air humidity, etc.

An important factor is a reduction in the insulating power of the chimney walls caused by changes in the physical properties of the insulation due to, e.g., the mineral wool getting damp [3]. As a result, the temperature of the flue gas in the chimney’s upper part decreases, and after the critical temperature is reached, this causes excessive flue gas condensation on the chimney lining.

As the condensate penetrates through cracks in the lining, it makes the mineral wool damp, whereby the latter loses its insulating properties. Moreover, the damp mineral wool sinks, and as a result, areas devoid of thermal insulation are created. As the coefficient of thermal conductivity decreases, the temperature of the chimney’s inner wall falls further, and so does the temperature of the flue gas. As a result, more and more flue gas condenses on the chimney’s inner wall, whereby the degradation processes in the chimney’s RC shell intensify (Figure 1, Figure 2 and Figure 3).

As condensation drips down the inner wall, it penetrates via cracks to the mineral wool and outside to the reinforcement of the concrete wall, causing intense corrosion of the concrete and the reinforcing steel (Figure 4).

## 3. The Idea of a Thermographic Survey of Chimney Thermal Insulation

Thermographic surveys [4,5,6,7,8,9] have been used to detect thermal bridges in residential buildings for many years. Thermographic surveys are conducted using a camera which can take thermograms. In the case of chimneys, only thermograms of the external surface of chimney’s wall are taken. The air temperature considerably affects the accuracy of thermographic surveys. The latter is more precise. The larger the thermal contrast between the surface of the chimney shell and the ambient air temperature the better the thermogram. Therefore, the best period for evaluating the condition of the thermal insulation in chimneys is in winter.

Passive thermography is the optimal method for the thermographic surveying of chimneys. Based on the thermographic images and reference readings, one can determine the temperatures on the surface of the chimney shell [3,10]. In this method, the image is obtained for a set scale range. Knowing the temperature of the flow gas flowing through the chimney and the temperature on the outer surface of the chimney shell, one can determine the actual thermal transmittance coefficient and compare it with its calculated values specified in the design documents. Based on the relative temperature differences, one can determine the degree of damage to the thermal insulation. The measurements were made by a company specializing in thermovision. The cameras were calibrated before measurements.

However, there may be a few percentages of error due to the curvature of the chimney surface, non-homogeneity of the surface, etc.

## 4. Thermographic Surveys of Chimney Thermal Insulation

The monitoring of the condition of the chimney’s insulation is an important element in the assessment of the durability of the chimney. Based on such monitoring, one can forecast the actual service lifespan of the structure and systematically eliminate the causes of chimney shell degradation. The condition of the thermal insulation of industrial chimneys is examined using the classic thermographic method [11,12]. It is mainly the structure’s envelope which is examined in this way to detect places where there are gaps in the insulation, or there is no insulation. In the considered case, the thermograms showed that temperature differences on the chimney’s outer surface occurred along practically its whole height and on its circumference (latitudinally). According to the results [13] of the thermographic surveys of the 120 m high chimney with the lower and upper diameter of respectively *D*_d_ = 7.16 m and *D*_g_ = 4.48 m, the temperature difference in the particular points of the surface amounted to

As much as 21.8 °C in the chimney’s upper part;15.9 °C in the chimney’s middle part;10.5–11.9 °C (being more uniform) in the chimney’s lower part (Figure 5).

The lower part of the chimney does not show such a high degree of degradation as its middle and upper parts. The thickness of the outer reinforced concrete shell, which is considerably greater in the lower part of the chimney, is one of the determining factors.

To nondestructively investigate the condition of the thermal insulation, an analysis of temperature values *v*_e_ on the outer surface of the chimney’s shell was carried out. The parameters for the analysis had been determined based on the known thermophysical characteristics of the materials, the chimney’s geometry (specified in the design documents), and temperature measurements. The following parameters of the layered chimney wall were assumed for the analysis:
Thermal conductivity:○fire brick = 1.30 [W/(m K)],○mineral wool = 0.05 [W/(m K)],○reinforced concrete = 1.74 [W/(m K)],Heat-insulating layer thickness:○fire brick layer—11.4 cm,○mineral wool layer—13 cm,○reinforced-concrete chimney shell wall—24 cm,Temperature:○inside chimney 200 °C,○external 12.3 °C.Temperature values *v*_e_ on the outer surface of the chimney shell are:○for the wall with mineral wool: *v*_e_ = 14.4 °C,○for the wall without a mineral wall: *v*_e_ = 32.4 °C.

Calculations and in situ measurements indicate that in the places where the temperature of the reinforced-concrete shell is the highest there is no mineral wool or the wool there has very poor insulating properties. The measured minimum temperatures of the shell in well-insulated places on average amounted to 17 °C while the maximum temperatures in the same thermograms on average amounted to 33 °C. The variation coefficient for the temperature on the chimney surface ranges from 0.13 to 0.25. The average decrease in thermal performance in the particular chimney wall surface areas ranges from 20% to 90%.

## 5. Thermal and Static Load Analysis

As a result of the nonuniform temperature distribution, caused by damage to the thermal insulation, additional internal forces, such as bending moments, shear stresses, and annular tensile and compressive forces, arise. The values of the forces can be traced by studying real cases of damaged chimneys.

### 5.1. Adopted Assumptions

The following were assumed: the velocity of flow of the flue gas in the chimney: *v* = 10 m/s, the operating temperature of the flue gas at the chimney outlet: *t*_w2_ = 180 °C, and the operating temperature of the flue gas at the chimney inlet: *t*_w1_ = 220 °C. The inlet of the combustion gas takes place through a connecting flue pipe located below the ground level.

The external design temperature was assumed based on the national annex to standard [14]: in winter *T*_min_ = −36 °C and in summer *T*_max_= + 40 °C.

Since no detailed information was available, the emergency flue gas temperature was assumed as 20% higher than the typical one. The emergency temperature amounts to *t*_wa,1_ = 220 °C and *t*_wa,2_ = 270 °C, respectively, at the chimney outlet and inlet. The emergency temperature was used in the calculations.

### 5.2. Types of Thermal Effects in the Chimney

The linearly elastic models of chimney concrete walls are the simplest models that indicate the influence of thermal factors. The wall thickness of the reinforced concrete chimney is not large as in the case of silos [15] for hot materials. The model of linear temperature gradient used for these silos gives excessive bending moments. The wall of a reinforced concrete chimney subjected to repeated wind and thermal loads behaves linearly. Another problem is with cracks in the wall which reduce the stiffness of the chimney. Cracks in a chimney reduce the impact of bending moments but also expose the chimney to corrosion. The first load results from the difference between the temperatures on the surface of the chimney shaft. The load was calculated for an ideal situation, i.e., immediately after chimney erection—no degraded insulation, and for a situation when some of the insulation has degraded. The temperature load was introduced as the gradient of the temperatures in the chimney shaft.

The second load stems from the difference between the chimney operating temperature during chimney service life and the initial temperature, i.e., the temperature at which the chimney was erected.

### 5.3. Distribution of Temperature in Chimney Wall for Undamaged Insulation

The chimney wall consists of the following layers: the shaft, the thermal insulation, and the lining. The thicknesses and diameters were assumed according to Table 1. As the reference situation, the condition of the chimney immediately after its erection, i.e., with the continuous mineral wool and fibre brick insulation, was adopted. The following thermal conductivity coefficients were assumed: λ_b_ = 1.74 W/mK for the reinforced concrete wall, λ_iz_ = 0.05 W/mK for the insulation (mineral wool), and λ_sz_ = 1.30 W/mK for the fire brick.

Thermal transmittance coefficient *k*:(1)$$\frac{1}{k}=\frac{1}{\alpha_{n}}+\underset{i}{\sum }\left(\frac{t_{i}}{\lambda_{i}}\kappa_{i}\frac{r_{z}}{r_{i}}\right)+\frac{1}{\alpha_{o}},$$ where $\alpha_{n}=8+v=8+10=18\;W/m^{2}K$—the inflow coefficient (the zone of the inner surface of the lining), where v—the mean velocity of the flue gas;


$\alpha_{o}=24\;W/m^{2}K$—the outflow coefficient (the outer surface of the shaft);$t_{i}$—the thickness of layer *i*;$\lambda_{i}$—the thermal conductivity coefficient of layer *i*;$r_{i}$—the outside radius of layer *i*;$r_{z}$—the outside radius of the shaft;$\kappa_{i}$—correction coefficients taking into account wall curvature:$\kappa_{i}={\left(\frac{r_{z}}{r_{i}}\right)}^{0.57}$ where i=b for the reinforced concrete shaft, i=iz for the mineral wool, i=sz for the fire brick.


The temperature drop in a given layer is expressed by the formula
(2)$$\Delta T_{i}=k\frac{t_{i}}{\lambda_{i}}\kappa_{i}\frac{r_{z}}{r_{i}}\Delta t,$$ where $\Delta t=t_{w}-t_{z}.$

For the inflow and outflow, the temperature drops are
(3)$$\Delta T_{n}=k\cdot \frac{1}{\alpha_{n}}\Delta t\;\Delta T_{o}=k\cdot \frac{1}{\alpha_{o}}\Delta t.$$

The temperature at the boundary of each of the layers is
(4)$$T_{j}=t_{w}-\frac{k}{\alpha_{n}}\cdot \Delta T-\underset{i}{\sum }\Delta T.$$

Figure 6 and Figure 7 show temperature drops in the particular layers in the winter season and in the summer season. The largest temperature drop occurs in the full insulation layer, and it is about 10 times larger than in the reinforced concrete shaft. Because of the small thickness of the fire brick layer, the temperature drop in this layer is the smallest.

### 5.4. Distribution of Temperature in Chimney Wall in Case of Insulation Discontinuity

Insulation discontinuities were assumed to occur in 2.5 high segments uniformly distributed along the whole height of the chimney. The mineral wool in the segments was assumed to lack the design insulating power.

The temperature drops in the particular layers for the decreased mineral wool insulating power are shown in Table 2. Since it is not possible to directly assess the degree to which the mineral wool’s insulating power decreased, an approximate method was used. Calculations in which $\lambda_{iz}$ was the unknown were carried out on the basis of the thermographic surveys of the existing chimney and the temperatures on its surface. The value of $\lambda_{iz}$ was adjusted consistently with the actual temperatures on the surface of the investigated chimney. The results are presented in Table 3. The results apply to the 10-fold decrease in the insulating power of the mineral wool in the selected places ($\lambda_{iz}=0.5\;W/\mathrm{mK}$).

The maximum temperature gradient in the reinforced concrete shell in the winter season increased by about 420% in comparison with the gradient calculated for the worst insulation case. For the summer season temperature, the gradient increased by about 405%. The temperature gradient which the insulation transfers is approximately equal to the gradient transferred by the chimney shaft under both the winter and summer temperature load. Such a large increase in temperature load can lead to the cracking of the reinforced concrete shell and to its damage. The temperature gradient values in the summer season and the temperature on the chimney’s surface in summer and in winter and a figure (Figure 8, Figure 9 and Figure 10) showing the temperature gradient values in each of the layers are presented below.

To relate the assumptions to the actual temperature loading of the chimney, the calculated gradients were compared with the temperatures appearing in the thermographic pictures of the existing chimney. The investigated real chimney is a 120 m high reinforced concrete structure serving a coke oven battery. The temperature of the combustion gas at the investigated chimney’s inlet reaches 270 °C (maximally 340 °C) while the flue gas temperature at the outlet amounts to 220 °C (maximally 300 °C). The chimney is made of concrete C25/30 and reinforced with steel A-II. The chimney was divided into 9 segments, each about 15 m high. The chimney’s insulation is made of semi-hard mineral wool boards and batts and its lining is made of fire brick. The thicknesses of the particular layers and their diameters are presented in Table 1.

The thermographic surveys revealed temperature differences on the chimney’s surface, which indicates that the insulation was not uniform, there were gaps in it, and in places the insulation had slid down and was damp. The temperature differences can also be due to the degradation of the reinforced concrete chimney shaft. Since the measurements were carried out outside the chimney, the places with deteriorated thermal insulation are visible as dark red (the hottest places). The suspected places where the insulation is missing are visible in the thermograms as bands running at regular intervals around the circumference of the chimney. One can discern a certain pattern in insulation discontinuity: the bands are spaced at every 1 to 3 m. The insulation loss is most visible in the lower part of the chimney, and it is uniform there. In the thermographic picture, the surface temperature in this part reaches about 34 °C. In the segments situated higher one can see more distinct temperature differences on the outside surface, ranging from 22 to 34 °C.

Since the temperature of the flue gas in the existing chimney is comparable with the assumed emergency temperature in the chimney and also the diameters and thicknesses of the layers are similar, the temperature on the surface of the RC shaft, indicated by the thermographic surveys should be similar to the one yielded by the calculations. As indicated by the high background temperature, the surveys were carried out in the summer season. The ambient temperature was assumed to be 15 °C. The temperatures on the shaft surface calculated for the case with insulation and the case with insulation loss are comparable with the ones revealed by the thermographic surveys of the real chimney, as shown in the tables. For the assumed value of $\lambda_{iz}=0.5\;W/\mathrm{mK}$, the temperature on the RC shell is comparable with the temperature indicated by the thermographic surveys.

Temperature loading in 2.5 m high bands for alternately full insulation and damaged insulation was assumed. The values were assumed according to Table 2 and Table 3.

### 5.5. Surface Temperature Load

Since the chimney works at a temperature different than the one prevailing during its erection, a temperature load generating internal forces due to the chimney’s limited freedom of deformation was assumed. Two load cases, the summer season load and the winter season load, were considered. In each of the cases, a load at full insulation and a load at damaged insulation were analysed.

The mean temperature in the wall can be calculated from the formula
(5)$$\bar{T}=T_{b,wew}-\frac{\Delta T_{b}}{2},$$ where $T_{b,wew}$—the temperature on the inside surface of the reinforced concrete shaft; $\Delta T_{b}$—the temperature drop in the reinforced concrete shaft.

The temperature at which the chimney had been erected was assumed as $T_{o}=10^{\;o}C$.

The difference between the chimney operating temperature and the chimney erection temperature was calculated from the formula

(6)$$\Delta T=\bar{T}-T_{o},$$

### 5.6. Increase in Bending Moments Due to the Increase in the Temperature Gradient

The difference in the temperature distribution across the thickness of the reinforced concrete wall between the case with damage insulation and the case with undamaged insulation amounts to 

*T* = 113 − 28 = 85°

Such a big gradient causes additional bending moment and additional shear force.

The model of these calculations is presented in Figure A1.

Calculations of this influence are included in Appendix A

Additional thermally induced internal forces.

Let us consider the chimney as a cylindrical shell in which the unexpandable (well thermally insulated) band can be regarded as an element resembling a clamping tendon. Circumference *L* of the surrounding area with damaged thermal insulation is increased by *L* due to higher temperature *T*. The tendon with the introduced compressive force reduces the circumference to its size before the thermal expansion.

Using such a static load model one gets additional bending moment *M* and additional shear force Q (model is presented on Figure A2. The calculation of these additional factors is presented in Appendix B.

A few remarks 

The presented computing models do not average their results. They assume quite a sharp transition from one temperature field to another without a transition area. In fact, the intermediate area can smooth calculations outcomes.

## 6. Conclusions Emerging from Investigations of Thermal Insulation of Chimneys

The method of thermographic surveys, used to monitor thermal insulation in housing, also finds an important application in the investigation of the durability and failure hazard of industrial structures in which the condensation of chemically aggressive gases is likely to occur. This particularly applies to chimneys where invasive tests are not preferred because of the consequences of taking samples from the structure.

Owing to thermographic surveys, one can monitor the hazards leading to the degradation of the chimney structure.

It is possible to indicate the area where flue gas condensate penetrates through cracks and leaks into the inner fire-brick lining into the insulation layer made of mineral wool. The condensate penetrating through cracks in the RC shaft to the structural reinforcement intensifies the corrosion of the steel and the concrete. As a result of the decrease in insulating power the temperature in the chimney RC core rises, whereby the bending moments and the tensile forces increase and the reinforced concrete shell cracks under the additional stresses which had not been taken into account in the design of the walls [2].

Thermograms can show increased shear and tensile stresses that can cause these large cracks in concrete.

Soaked by condensate or damaged mineral wool areas are visible on thermograms.

The calculations presented in the paper indicate the possibility of assessing the values of shear stresses from thermal interactions.

The authors took part in the design of chimney repairs with large diagonal cracks caused by high shear stresses.

In the case of chimneys operating in the uninterruptable process mode, special measures are required to improve the condition of the insulation. The repair of the reinforced concrete chimney consisting solely in injection filling the external cracks [10] stops the corrosion of the reinforcing steel, but it does not improve the insulation of the chimney.

## Figures and Tables

**Figure 1 materials-12-02027-f001:**
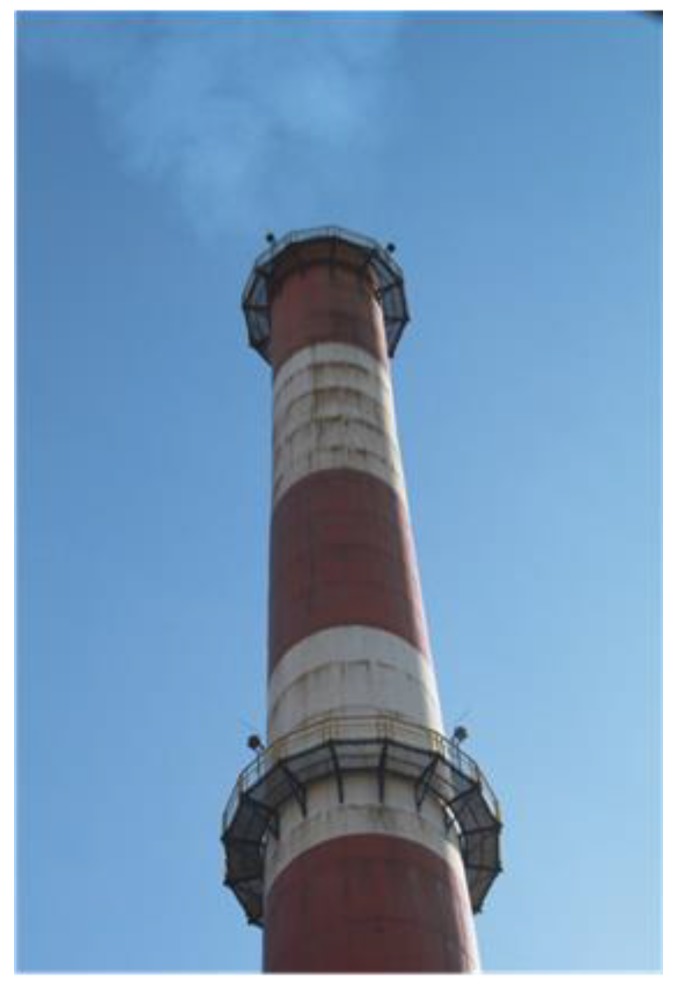
Deposition of condensation on outer surface chimney wall along construction joints.

**Figure 2 materials-12-02027-f002:**
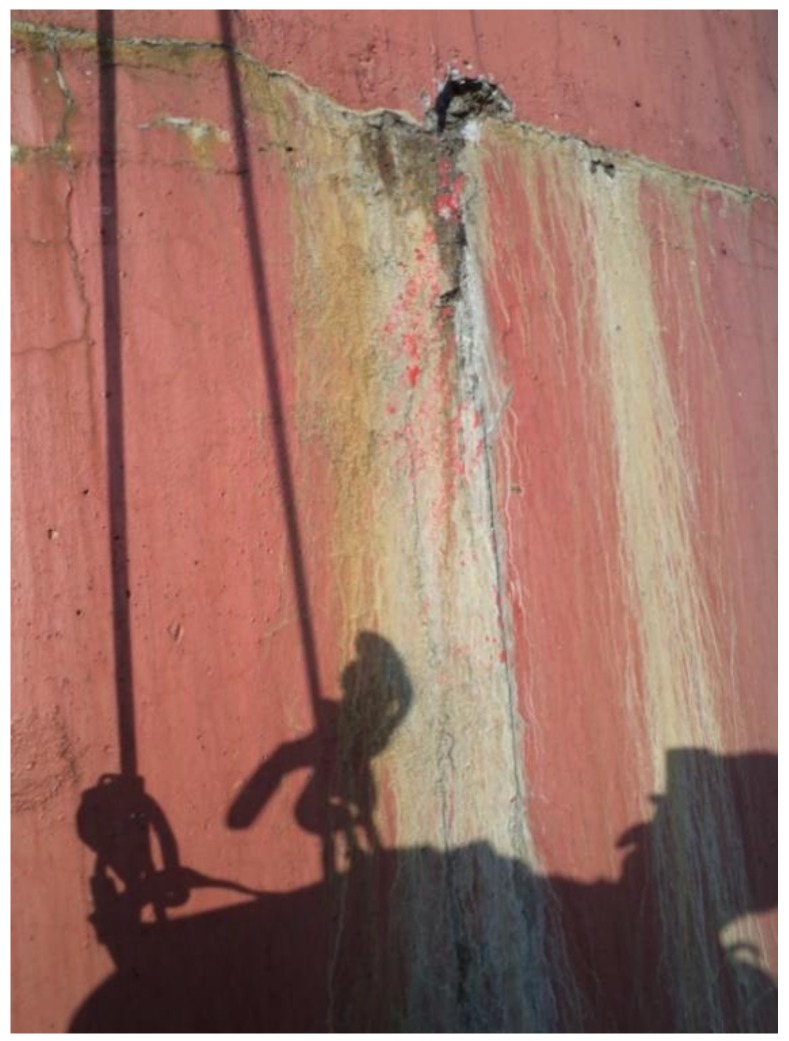
Condensation degrading of chimney wall.

**Figure 3 materials-12-02027-f003:**
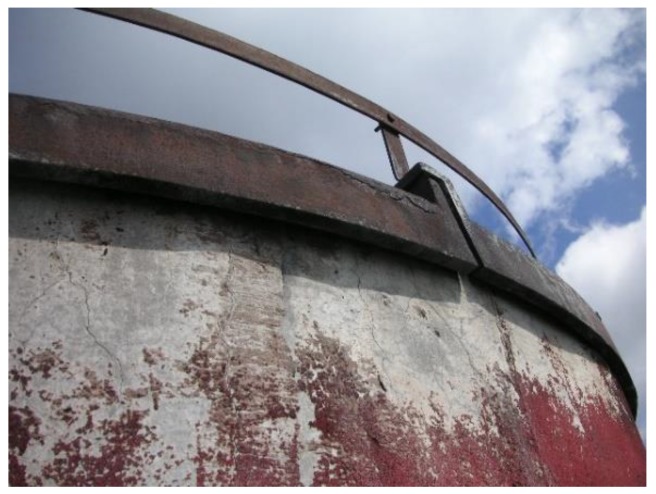
Condensation dripping down from a chimney upper ring.

**Figure 4 materials-12-02027-f004:**
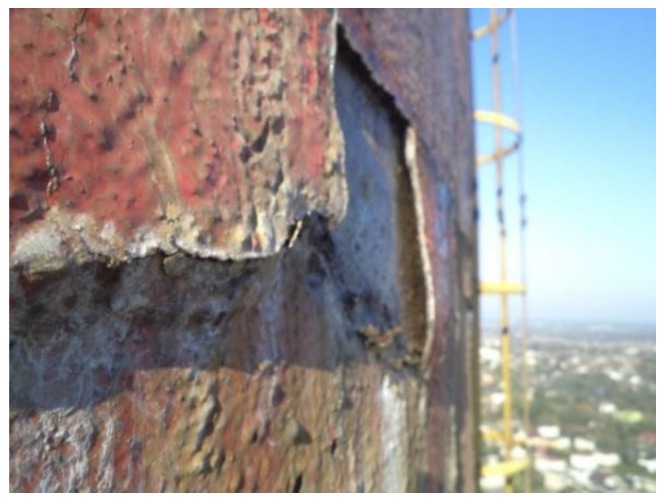
Condensation penetrating chimney crown wall.

**Figure 5 materials-12-02027-f005:**
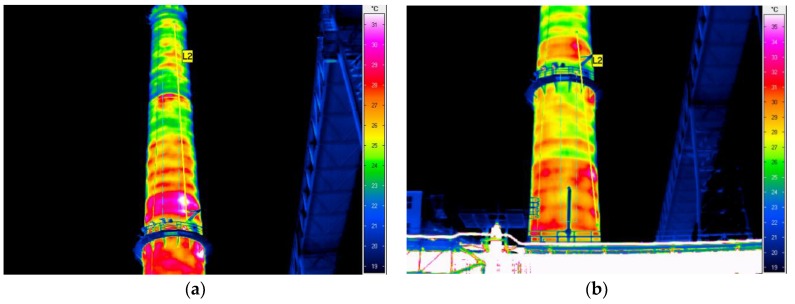
Chimney thermograms revealing insulation imperfections: (**a**) No thermal insulation more than two metres above and under platform; (**b**) Varied temperature distribution in chimney shaft.

**Figure 6 materials-12-02027-f006:**
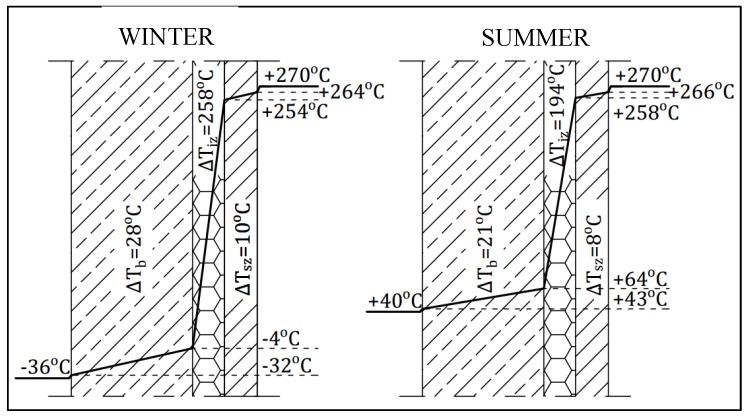
Temperature drops in particular layers for the winter and summer season.

**Figure 7 materials-12-02027-f007:**
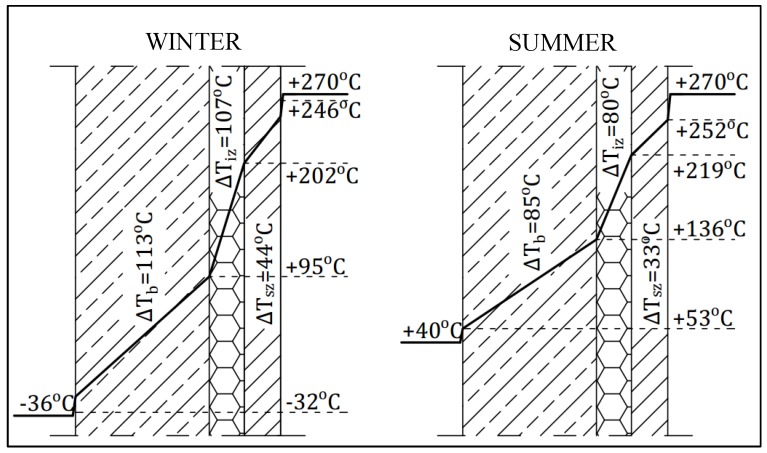
Temperature drops in particular layers of design chimney insulation in the first segment with damaged insulation.

**Figure 8 materials-12-02027-f008:**
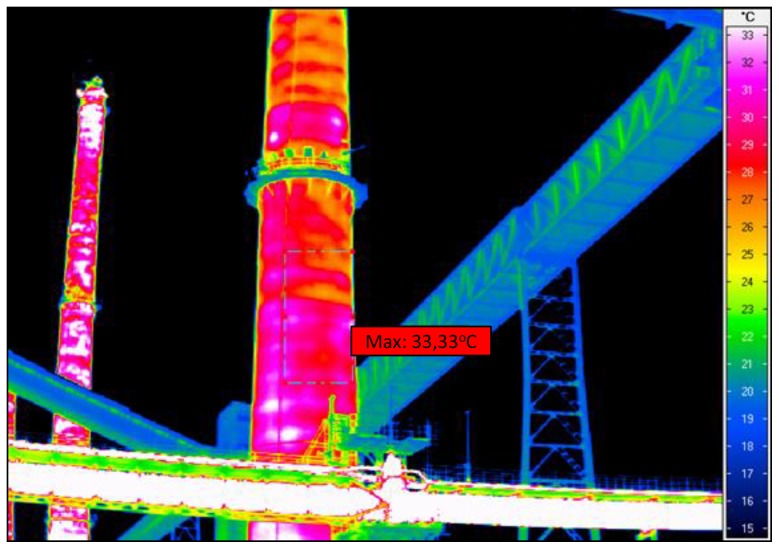
Thermographic picture of the lower part of chimney shaft.

**Figure 9 materials-12-02027-f009:**
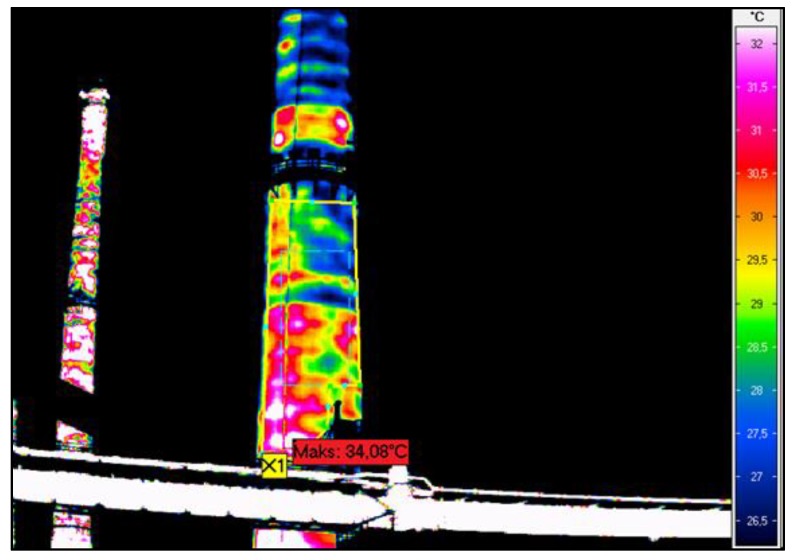
Thermographic picture of the middle part of chimney shaft.

**Figure 10 materials-12-02027-f010:**
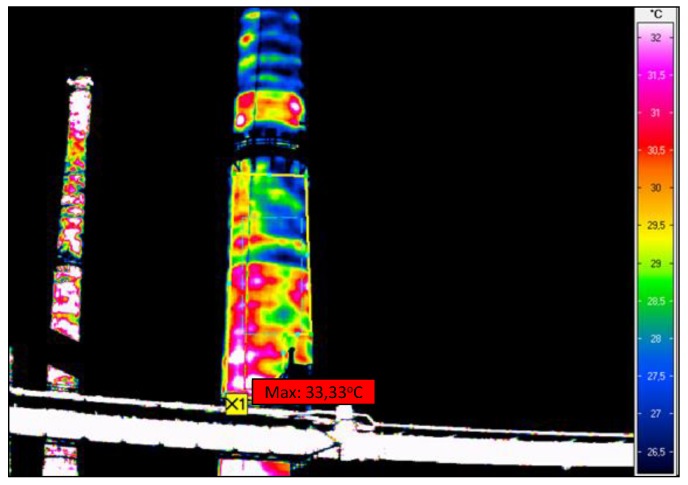
Thermographic picture of the upper part of chimney shaft.

**Table 1 materials-12-02027-t001:** Geometric dimensions of the thermographically surveyed existing chimney.

Segment No.	H	Shaft Outside Radius	Chimney Wall Thickness	Shaft Inside Radius	Insulation Thickness	Lining Thickness
-	[m]	[m]	[m]	[m]	[m]	[m]
1	2.5	4.00	0.42	3.58	0.15	0.23
2	15	3.80	0.39	3.41	0.15	0.23
3	30	3.61	0.36	3.25	0.13	0.23
4	45	3.41	0.33	3.08	0.13	0.114
5	60	3.21	0.3	2.91	0.13	0.114
6	75	3.01	0.27	2.74	0.13	0.114
7	90	2.82	0.24	2.58	0.13	0.114
8	105	2.62	0.21	2.41	0.13	0.114
9	120	2.42	0.18	2.24	0.13	0.114

**Table 2 materials-12-02027-t002:** Temperatures on the surface of the calculated chimney at an ambient temperature of 15 °C.

**Seg. No.**	**H**	**Damaged Insulation**	***T*** **_sz,w_**	***T*** **_sz,z_**	***T*** **_iz,z_**	***T*** **_b,z_**	***T*** **_ext._**	**Full Insulation**	***T*** **_sz,w_**	***T*** **_sz,z_**	***T*** **_iz,z_**	***T*** **_b,z_**	***T*** **_ext._**
-	[m]		[°C]	[°C]	[°C]	[°C]	[°C]		[°C]	[°C]	[°C]	[°C]	[°C]
1	10		250	213	124	30	15		265	256	41	19	15
2	20		246	208	118	30	15		261	252	39	19	15
3	30		241	203	112	30	15		257	248	38	19	15
4	40		234	192	118	32	15		251	240	42	20	15
5	50		230	187	111	33	15		247	236	40	20	15
6	60		223	175	113	35	15		241	227	44	21	15
7	70		218	169	106	35	15		237	223	41	21	15
8	80		212	160	103	37	15		232	216	42	22	15
9	90		207	153	95	37	15		228	212	39	22	15
10	100		204	150	94	37	15		224	208	38	21	15
11	110		200	148	92	36	15		220	204	38	21	15
12	120		193	142	89	35	15		212	197	37	21	15

Where *T*_sz,w_—the temperature of the outside surface of the ceramic wall; *T*_sz,z_—the temperature of the inside surface of the ceramic wall (contact with isolation); *T*_iz,z_—the temperature of the inside surface of the concrete wall; *T*_b,z_—the temperature of the outside surface of the concrete wall; *T*_ext_—outside temperature.

**Table 3 materials-12-02027-t003:** Differences between chimney operating temperature and chimney erection temperature.

Seg. No.		H	Winter	Summer	
-	[m]	Insulation		Insulation	
		Full	Damaged	Full	Damaged
		[°C]	[°C]	[°C]	[°C]
1	10	−28	28	44	86
2	20	−29	25	43	83
3	30	−30	22	42	81
4	40	−27	27	44	84
5	50	−28	23	43	81
6	60	−25	26	45	83
7	70	−27	22	44	79
8	80	−25	21	45	79
9	90	−27	17	44	75
10	100	−28	16	43	74
11	110	−28	15	43	73
12	120	−29	13	42	71

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
