# Peer review of "Non-Destructive Testing of Technical Conditions of RC Industrial Tall Chimneys Subjected to High Temperature"

_materials, 2019, doi:10.3390/ma12122027_

Round 1

Reviewer 1 Report

In this study, the authors provide sufficient background to show the importance of the study. They used a thermographic camera to measure the temperature map of a real industrial chimney. Then, they used theoretical equations to calculate the temperature gradient in concrete, insulation, and brick. However, I could see the specific conclusions that were drawn in this study. So, I suggest the authors to clearly state the conclusions.

The Abstract provide too much background. You should tell the readers the research methods used in this study and show what you found in this study. The same problem for the Conclusions. You should summarize the results rather than state the research background.

The boundary condition for calculations in Section 5 is not realistic. When the concrete shell gets hot, air close to the shell is heated up as well. This means that temperature at the boundary is higher than the external air temperature. Therefore, the temperature gradient in the concrete shell is not as sharp as shown in this study.  The thermal damage to concrete can not be so serious anymore.

Author Response

1.            Abstract is corrected and we included the appropriate annotations

2.            Conclusions are completed  and more suitable to the whole article.

3.            The measurements were made by a company specializing in thermovision.   The cameras were calibrated before measurements. However, there may be a few percentage error due to the curvature of the chimney surface, non-homogeneity of the surface, etc.(line 99 - 102)

4.            The linearly elastic models of chimney concrete walls   are the simplest models that indicate the influence of thermal factors. The wall thickness of the reinforced concrete chimney is not large as in the case of silos [ 15] for hot materials. The model of linear temperature gradient used for these silos gives excessive bending moments. The wall of a reinforced concrete chimney subjected to repeated wind and thermal loads behaves linearly. Another problem is with cracks  in the wall which  reduce the stiffness of the chimney. Cracks  in chimney reduce the impact of bending moments but also expose the chimney to corrosion. The first load results from the difference between the temperatures on the surface of the chimney shaft. The load was calculated for an ideal situation, i.e. immediately after chimney erection – no degraded insulation, and for a situation when some of the insulation has degraded. The temperature load was introduced as the gradient of the temperatures in the chimney shaft.(line 168 - 178)

5.            All indicated errors are corrected

Reviewer 2 Report

A method is proposed to practically evaluate The, degradation level in industrial chimneys. The process and the suggested method are straightforward and seems efficient. So, I consider it acceptable for publication with minor revision. The following concerns must be addressed in order to have a satisfactory draft for online publication.

-        First, the abstract does not address two point. One is to concisely address the core method used for evaluation of damage. And the abstract does not conclude the paper.

-        Please explain the effect of paintings, dusts, and dirts on the temperature calculations. Is it considered that surface painting can affect the real temperature calculations?

-        There are a lot of other factor rather than linear elastic models that can change the predicted model. For example the prestressed beams have different stress distribution and do not necessarily go to yielding point by temperature difference. Do the authors consider this? It would be nice if authors explain it in the text. Also the plastic behavior of different materials in the structure can definitely complicate the whole idea behind the depredation estimation. Please clarify in text.

Some minor issues:

-        Minor English issues are seen, but negligible. Line 30, delete one of the ‘often’s. Line 82-83 the sentence is not understandable in my perspective.

-        Please make sure all the figures and tables contain the units. Table 3 lakes enough units.

-        Table 1, diameter is the product of radius (D=2R). Why does it need to report both numbers?

-         Table 2, the temperatures are not referred in text. What is T##,#? please define the variables in the text.

I believe this can be a decent publication, if authors precisely address all of the above issues.

Thank you.

Author Response

1.     Abstract is corrected and we included the appropriate annotations

2.     The measurements were made by a company specializing in thermovision.   A laboratory worker compensated on a thermal camera  by introducing to this devise real parameter of humidity, ambient temperature, distance to the object, etc. But we must have own correction

3.     We will be able to correlate the temperature measured by camera with  the temperature of the surface itself in the future

4.     However, there may be a few percentage error due to the curvature of the chimney surface, non-homogeneity of the surface, etc.(line 99 - 102)

5.     During the measurement, we focused on the temperature differences between the neighboring  areas because gradient of temperature across and along the chimney wall causes the maximum values of stresses, which we sought.

Reviewer 3 Report

The paper is generally well-written; although there is definitely room to improve the English writing. I have a major concern regarding the actual contribution of the paper. As the author mentioned, the method has been applied to residential building for decades. Basically, for any types of structure with thermal insulation issues, it’s common to use thermographic survey. So, I don’t see any contribution in this regards which is necessary for research paper. Additionally, the way, the paper presented does not establish the ground for research paper. It is just a case study that can be done by a consulting firm particularly Appendix A. However, I cannot deny the value of the method in this application assuming a step by step procedure is presented to the readers. I suggest the authors have a second thought about the format of the paper, maybe first establish the framework for thermal non-destructive testing of tall reinforced concrete chimneys and then talk about the case study and implement the framework discussed earlier in the actual real-life example. In addition, the comments below can help improving the quality as well: - There is no actual literature review. The authors should do an extensive literature review on the subject and distinguish the gap in the body of knowledge and then establish what they are going to contribute. - A few sentences should be added to the abstract as well as introduction to establish the main contributions of the paper. - Using bullets in the research paper is not common e.g. line 46, 47 , 49, 59, 60,61 and etc. - The title of the article is somehow deceiving. I suggest to add a word thermal to clarify the scope of the paper; also what does “high” mean? Are you referring to tall chimneys or highly reinforced chimneys? I think you meant the later. Revise. - In Figure 8 and 9, I assume you are referring to “max” by “Maks” correct. - The structural aspect of the problem is widely neglected; which I believe, is the fundamental. I suggest define the scope of work vividly for the work.

Author Response

1.     Abstract is corrected and we included the appropriate annotations

2.     Conclusions are completed  and more suitable to the whole article.

3.       We changed title of this article to more appropriated to the content in the article.

4.       We have completed the bibliographies

5.       WE corrected some language mistakes

Thank you for your remarks

Round 2

Reviewer 1 Report

Even though the authors improved the abstract, it is still not clear for me: the work done by this study and the main conclusion. Sentences like "Acid condensate stored in mineral wool destroys the concrete cover" are the authors' subjective assumption but not being proved by this study. Words like "should" and "one can monitor" indicate the suggestions but actually, they are not. As stated by the authors, the measurements were made by a company. Therefore, the data analysis by the authors should be mentioned in the abstract. 

Other comments:

1. The sentence "The reduction of chimney insulation, the soaking of insulating mineral wool by condensate leads ..." needs to be revised.

2. "should monitored" -> "should be monitored"

3. The repetition of "Thermographic surveys are conducted using a camera which can take thermograms.

There are other mistakes which should be carefully checked and revised. 

Author Response

Answers  of Authors of article to the comments of the Reviewer

The abstract has been simplified and condensed

Errors have been corrected

Thank you very much

MMaj

AUbysz

Reviewer 3 Report

The authors did not apply my comment to consider having a second thought about the format of the paper, maybe first establish the framework for thermal non-destructive testing of tall reinforced concrete chimneys and then talk about the case study and implement the framework discussed earlier in the actual real-life example. Anyway, I would accept it in the current format. 

Author Response

(The authors gave the same response as above.)
